# Comprehensive Investigation of Angiogenesis, PASS Score and Immunohistochemical Factors in Risk Assessment of Malignancy for Paraganglioma and Pheochromocytoma

**DOI:** 10.3390/diagnostics14080849

**Published:** 2024-04-19

**Authors:** Marija Milinkovic, Ivan Soldatovic, Vladan Zivaljevic, Vesna Bozic, Maja Zivotic, Svetislav Tatic, Dusko Dundjerovic

**Affiliations:** 1Department of Pathology, University Clinical Center of Serbia, 11000 Belgrade, Serbia; vesna.bozic.ph@gmail.com; 2Institute for Medical Statistics and Informatics, Faculty of Medicine, University of Belgrade, 11000 Belgrade, Serbia; soldatovic.ivan@gmail.com; 3Clinic for Endocrine Surgery, University Clinical Center of Serbia, 11000 Belgrade, Serbia; vladanzivaljevic@gmail.com; 4Institute of Pathology, Faculty of Medicine, University of Belgrade, 11000 Belgrade, Serbia; majajoker@gmail.com (M.Z.); svetislav.tatic@med.bg.ac.rs (S.T.); drdundjerovic@gmail.com (D.D.)

**Keywords:** angiogenesis, pheochomocytoma, paraganglioma, CD105, ERG, PASS, Ki67, S100, SDHB

## Abstract

A challenging task in routine practice is finding the distinction between benign and malignant paragangliomas and pheochromocytomas. The aim of this study is to conduct a comparative analysis of angiogenesis by assessing intratumoral microvascular density (MVD) with immunohistochemical (IHC) markers (CD31, CD34, CD105, ERG), and S100 immunoreactivity, Ki67 proliferative index, succinate dehydrogenase B (SDHB) expressiveness, tumor size with one the most utilized score Pheochromocytoma of Adrenal Gland Scales Score (PASS), using tissue microarray (TMA) with 115 tumor samples, 61 benign (PASS < 4) and 54 potentially malignant (PASS ≥ 4). We found no notable difference between intratumoral MVD and potentially malignant behavior. The group of potentially malignant tumors is significantly larger in size, has lower intratumoral MVD, and a decreased number of S100 labeled sustentacular cells. Both groups have low proliferative activity (mean Ki67 is 1.02 and 1.22, respectively). Most tumors maintain SDHB expression, only 6 cases (5.2%) showed a loss of expression (4 of them in PASS < 4 group and 2 in PASS ≥ 4). PASS score is easily available for assessment and complemented with markers of biological behavior to complete the risk stratification algorithm. Size is directly related to PASS score and malignancy. Intratumoral MVD is extensively developed but it is not crucial in evaluating the malignant potential.

## 1. Introduction

Paragangliomas are rare non-epithelial neuroendocrine tumors, but potentially life-threatening. Several classifications and terms have been used for these tumors, depending on the place of origin, like the term pheochromocytoma for ones that originate from the medulla of the adrenal gland and paraganglioma for tumors arising in other anatomical places of the autonomic nervous system [1]. Pheochromocytomas are according to the new nomenclature intra-adrenal paragangliomas, and classic sympathetic tumors that produce and release catecholamines: noradrenaline and/or adrenaline [1,2]. Other paragangliomas are extra-adrenal and they can be parasympathetic and sympathetic. Parasympathetic paragangliomas are primarily located in the head and neck (so-called head and neck paraganglioma-HNPGL) and do not release catecholamines, while sympathetic paragangliomas are located in the chest, abdomen, and pelvis, and release large amounts of noradrenaline, so like pheochromocytomas, they are accompanied by symptoms such as hypertension, palpitations, sweating, headaches and diabetes mellitus [2]. These non-epithelial neuroendocrine tumors can occur at any age, with prevalence in the fourth and fifth decade, and they occur equally in men and women [3]. The only known causative factor for paragangliomas is hereditary susceptibility, and for these tumors, genetic contribution is the strongest [3,4,5].

Apart from the clinical symptoms that put at risk the patient, paragangliomas can have a benign clinical course. Ideally, they can be localized, and surgery, as the primary form of therapy, can be a successful and definitive form of therapy [6]. But in the case of malignant behavior and with excessive release of catecholamines, there is no generally effective therapy [7,8]. The prevalence of malignancy is 5% to 26% of cases [9]. It is extremely difficult at the time of diagnosis to predict whether paraganglioma will have a malignant or benign clinical course, especially since metastases can occur much later after the primary tumor [10].

In the Classification of Tumors of Endocrine Organs of the World Health Organization (WHO) from 2017, in the absence of clearly defined malignancy criteria, it is considered that all pheochromocytomas and paragangliomas can potentially metastasize, and instead of classifying them as benign and malignant, it is preferable to stratify the risk of metastasis in these tumors [3]. These recommendations are retained in the latest WHO classification from 2022 [1]. Although it was challenging, a staging system for intra-adrenal and extra-adrenal sympathetic paragangliomas was introduced in 2017 in the 8th Edition AJCC (the American Joint Committee on Cancer) Cancer Staging Manual [3].

Numerous risk factors have been analyzed like localization, and it was observed that 10% of parasympathetic paragangliomas metastasize, up to 25% of pheochromocytoma cases, and 40–70% of sympathetic paraganglioma cases [1,11,12,13,14].

Tumor size and weight showed some correlation with metastatic potential but not as independent variables [11].

Essential histological characteristics that could indicate malignant potential were analyzed and included in the Pheochromocytoma of the Adrenal gland Scaled Score (PASS) by Thompson, who made this scoring system [15]. Although its use remains controversial it is a commonly used scoring system [16,17,18]. PASS scoring system includes twelve histological characteristics: capsular invasion, vascular invasion, extension into the periadrenal adipose tissue, presence of large nests or diffuse growth (in >10% of tumor volume), central tumor or confluent necrosis, high cellularity, tumor cell spindling, cellular monotony, increased mitotic figures (>3/10 high power field), atypical mitotic figures, profound nuclear polymorphism, and nuclear hyperchromasia. The maximum score is 20. The cutoff value is 4, so when the sum is <4 it is considered that the tumor will have a benign course, while a score ≥ 4, means that the tumor is likely histologically more aggressive and potentially malignant.

Angiogenesis is substantial for tumor growth and metastases so it has been explored in many studies, with different approaches including assessing microvascular density (MVD), vascular patterns, and expression of angiogenic factors but with inconclusive results [19,20].

Ancillary use of many immunohistochemical markers showed some significance and association with the malignant potential of these tumors, such as the proliferative index Ki67, S100 to determine the number of sustentacular cells, and succinate dehydrogenase B (SDHB) for screening patients with hereditary paraganglioma pheochromocytoma syndrome (loss of expression of SDHB may indicate malignant biological potential) [1,14,21].

The primary objective of our research is to further explore angiogenesis, determining intratumoral MVD using IHC endothelial markers CD31, CD34, CD105, and nuclear endothelial marker ERG (Erythroblast transformation-specific(ETS)- Related Gene) and then look into how intratumoral MVD can contribute to risk stratification by comparing them with histological characteristics using the PASS scoring system. Furthermore, the aim is to make a comprehensive analysis of known markers like immunohistochemical expressions of Ki67, S100, and SDHB, along with the tumor’s size and weight. The application of TNM staging (The TNM Staging System includes the extent of the tumor (T), extent of spread to the lymph nodes (N), and presence of metastasis (M)), in a retrospective manner was undertaken to ensure a comprehensive understanding of the clinical context and facilitate future analyses [22].

## 2. Materials and Methods

### 2.1. Study Population

We retrospectively analyzed patients who underwent surgery, due to paraganglioma, at the University Clinical Center of Serbia in Belgrade in the period from the end of 2008 to the beginning of 2018. In the study, we included all cases in which there was a consensus on the diagnosis mentioned above, as two pathologists independently performed examination (revision) of all selected cases. Among 110 consecutive patients who were operated on because of tumors, five of them were operated on twice during this period (four because of bilateral pheochromocytoma, and one had recurrence). In summary, we gathered 115 paraffin-embedded tumor tissue samples from 110 consecutive patients.

### 2.2. Tissue Microarray Construction

To identify representative areas in the donor paraffin-embedded tissue block, we utilized hematoxylin and eosin-stained (H&E) four-micron sections of each tumor. We manually punched tissue cores from the selected areas and arranged them on a recipient paraffin block. All the patients are represented with at least one representative core tissue cylinder. 

### 2.3. Immunohistochemistry

A tissue microarray was made on which immunohistochemical analysis was performed [23]. There were a total of 107 intra-adrenal paragangliomas with 1 recurrent case and 7 extra-adrenal paragangliomas. We cut 4 µm thick sections from the tissue microarray paraffin blocks using a tissue microtome. These sections were then subjected to deparaffinization and antigen unmasking procedures before being labeled with primary antibodies against CD31, CD105, CD34, ERG, SDHB, S-100, and Ki67. The list of used antibodies and manufacturers is shown in Table 1.

As a control for immunohistochemical staining, external tissue controls were used for all the mentioned antibodies. Immunohistochemical staining was performed manually, following the manufacturer’s instructions for each antibody separately [24]. Visualization of the immunohistochemical reaction was performed using the streptavidin-biotin technique (DAKO LSAB+ kit).

### 2.4. Morphometry

For further analysis, the IHC stained slide preparations were photographed with a digital camera Olympus DP70 (Olympus Corporation; Tokyo, Japan) through an upright microscope (Olympus BX50, Olympus Corporation) at a magnification of 200× times, creating images at 300 × 300 resolution, (photo dimensions 4080 × 3072 pixels). The digitalization of all slides was additionally executed with the Leica Aperio AT2 slide scanner (Leica Biosystems, Nussloch GmbH, Nußloch, Germany) for analysis and documentation purposes. Virtual slides generated from Leica Aperio AT2 were morphometrically analyzed with a Leica AperioImageScope (version 12.4.6, Leica Biosystems, Nussloch GmbH, Germany) and with FIJI-ImageJ2 software 2.3.1 [25].

All tissue samples are then divided into two categories according to the PASS score (PASS < 4 and PASS ≥ 4).

We counted sustentacular cells labeled with S 100 antibody on a two-dimensional section in a created network over the image. Cells whose positive nuclei are located in cubes, and cell extensions and nuclei that cross the upper and right border, are included and those that cross the left and lower border are excluded [26,27]. The average number of sustentacular cells per 1 mm^2^ was calculated according to the measured tumor surface. Proliferation index Ki67 was determined on each tissue cylinder by the percentage of positive tumor nuclei which are counted using the Cell Counter option. We noticed granular cytoplasmic staining in tumor cells as positive for SDHB, and the absence of staining in tumor cells with positive endothelial cells, as negative and loss of immunohistochemical SDHB expression.

### 2.5. Determination of Intratumoral MVD

We analyzed endothelial cytoplasmic and/or membrane markers of angiogenesis CD31, CD34, and CD105 and counted blood vessels according to the authors Weidner and Tanigawa, where each individual or group of endothelial cells that are clearly separated from adjacent microvessels, tumor cells, or other connective tissue elements is counted as a single blood microvessel. Vessel lumen, although often present, is not necessary to define a structure as a microvessel. Every 40 µm of microvessel length was counted as a new microvessel [28,29,30]. In addition to cytoplasmic/or membrane endothelial markers, we also counted blood vessels using the nuclear marker ERG. In the photo of each cylinder, using the Cell Counter option, we counted blood vessels and displayed intratumoral MVD as an average number per 1 mm^2^.

### 2.6. Clinical Data

Measurements of blood pressure, catecholamines, and/or their metabolites in plasma or urine, imaging data (radiology and nuclear medicine), genetic testing data, and macroscopic findings (tumor size and weight) were obtained from medical records, with the approval of Ethical committees.

### 2.7. Statistical Analysis

We employed both parametric (*t*-test) and non-parametric (Chi-square test, Mann–Whitney U test) difference tests, along with Spearman’s correlation analysis as a correlation test. The selected significance level, which represents the probability of the first type of error, is 0.05.

## 3. Results

### 3.1. Clinical and Macroscopic Findings

The research was conducted on 115 tumor tissues, 75 (65.2%) females, and 40 (34.8%) males. All cases underwent preoperative radiological imaging with a CT (computed tomography) scan, in some it was supplemented with magnetic resonance imaging. Elevated arterial pressure and/or paroxysmal pressure spikes were recorded in 91 (79.8%) patients. Among the total of 109 patients (accounting for 94.8%), pertinent information regarding the analysis of catecholamines and/or their metabolites in plasma or urine was available, and of these, 93 patients (representing 85.3%) exhibited elevated values. Amongst the cohort of 52 patients (constituting 45.2%), a supplementary diagnostic metaiodobenzylguanidine (MIBG) scintigraphy was conducted, revealing an augmented accumulation in the tumor region for 39 of the cases (equating to 75%). For 47 tissue samples (40.83%) information about genetic testing was available. Seventeen were confirmed to have Multiple Endocrine Neoplasia Type 2a (MEN2a), 10 patients had Von Hippel–Lindau Disease (VHL), and four patients had Von Recklinghausen Disease (also known as Neurofibromatosis Type 1). In this group, only one patient was confirmed to have an SDHB mutation. At the time of diagnosis and surgery, 64 (55.7%) were over 50 years old, and 51 (44.3%) were under 50 years old. Clinicopathological data according to the PASS score, are shown in Table 2.

In 60 (52.2%) cases, tumors were ≥50 mm, and in 55 (47.8%) cases, they were <50 mm. The mass of the tumor was greater than 60 g in 34 (29.6%) cases and less than 60 g in 81 (70.4%) cases. Comparing tumor size (in mm) and PASS values our research has shown that there is a significant association (*p* < 0.001) with medium strong correlation (r = 0.431).

### 3.2. Microscopic and Immunohistochemical Findings

Intratumoral MVD was examined as previously described for the endothelial nuclear marker (ERG) and for endothelial cytoplasmic markers (CD34, CD105, CD31) and compared with group PASS < 4 benign and PASS ≥ 4 malignant (Figure 1 and Table 3). Staining for each endogenous marker is shown in Figure 2. No significant statistical difference in average intratumoral MVD in group PASS < 4 benign and PASS ≥ 4 malignant was found, (ERG *p* = 0.071, CD34 *p* = 0.077, CD105 *p* = 0.088, and CD31 *p* = 0.337). Average intratumoral MVD was lower in the malignant group, for all four antibodies. Comparing average intratumoral MVD and PASS score, there was no correlation between average intratumoral MVD and potential malignant behavior (ERG *p* = 0.590, CD34 *p* = 0.213, CD105 *p* = 0.139, and CD31 *p* = 0.849). Intratumoral MVD, labeled with ERG, also has no considerable correlation with tumor size (mm) and tumor weight (g), *p* = 0.752 and *p* = 0.786, respectively.

Sustentacular cells were labeled with S100 antibody and shown in Figure 3. In our study, we counted the total number of sustentacular cells and expressed it as an average number per mm^2^ of tumor tissue surface (Figure 4). The average number of sustentacular cells per mm^2^ is 203.43, and we found no statistically notable correlation (*p* = 0.080). In the potentially benign (61 cases) group, the average number of sustentacular cells is 243, and in the potentially malignant (54 cases) group, that number was lower, 158.61. The number of sustentacular cells is decreased in the group of PASS ≥ 4 tumors but without a statistically meaningful difference (*p* = 0.062) (Table 3).

The proliferative index Ki67 was determined as described and the values shown in relation to PASS (PASS < 4 and PASS ≥ 4) (Figure 5). In our study, the PASS ≥ 4 group, the potentially malignant had a mean Ki67 of 1.22% and the potentially benign group (PASS < 4) had a mean Ki67 of 1.02%, with no statistically significant difference in these two groups (*p* = 0.598) (Table 3). Examining the relationship between values of Ki67 and PASS scores no significant correlation was found (*p* = 0.114). Detailed analysis by different values of Ki67 showed among 54 cases in PASS ≥ 4 group, 10 (18.5%) had a Ki67 value of ≥1%, 6 (11.1%) had a Ki67 value of ≥2% and 4 (7.4%) had a Ki67 value of >3%. The remaining 34 (62.96%) cases in group PASS ≥ 4 had a Ki67 value < 1%. Among the 61 cases in the PASS < 4 group, 39 (69.9%) had a Ki67 value of <1%, and 53 (86.9%) had a Ki67 value of <2%. However, 7 (11.5%) cases had a Ki67 value of ≥2%, 14 (22.9%) had a Ki67 value of ≥1%, and 1 case had a Ki67 value of >3% (1.64%) with the same conclusion, changing the cutoff value of Ki67 does not change differences in these two groups (PASS < 4 and PASS ≥ 4).

We examined the SDHB immunohistochemical expression, in tumor cells, some cases were shown in Figure 6 and Figure 7. In our study 109 cases (94.8%) were positive and 6 cases (5.2%), lost immunoreactivity. In the PASS < 4 group 4 cases (6.6%) were SDHB negative and 57 cases (93.4%) were SDHB positive, while in PASS ≥ 4 group 2 cases (3.7%) were negative and 52 (96.3%) were positive. There was no notable difference in losing SDHB immunoreactivity among group PASS < 4 and PASS ≥ 4 (*p* = 0.683) (Table 3).

Some patients underwent genetic testing, so data were available for 47 tissue samples (40.87%), with 32 (27.83%) of them having a confirmed mutation, data are summarized in Table 4.

The mean number of ERG-labeled blood vessels in the group of patients with a confirmed mutation was 277.15, whereas in the group without the mutation, it was 266.65. Our analysis showed that the difference in intratumoral MVD was not statistically significant between these two groups (*p* = 0.583).

## 4. Discussion

The primary difficulty particularly with these non-epithelial neuroendocrine tumors is that there are no established standards for determining their malignancy, and patients must undergo long-term monitoring and face uncertainty since metastases can develop even many years after the initial tumor [10,31,32]. The primary objective during histological examination following surgery is to accurately diagnose a malignant tumor that has not yet spread to other parts of the body. Numerous studies have been conducted, and efforts are being made to develop an algorithmic model based on known factors that indicate malignancy. In many studies, the PASS score is a reliable prognostic factor, even though the evaluation of certain histological parameters such as hyperchromasia, nuclear pleomorphism, and cellularity are subjective. Tumors with a PASS score less than 4 did not develop metastases, indicating that only patients with a PASS score of 4 or higher would require follow-up, including annual radiological imaging and biochemical measurements of metanephrines [18]. In recent assessments of the PASS, as well as other scoring systems that have been introduced since its development, the WHO Classification of Endocrine Tumors recognizes that while these systems have their advantages and disadvantages, their use should not be discouraged in further research, but instead should be optimized in conjunction with molecular analyses [1]. Considering this, we employed the PASS score as a suitable standard for our diagnostic algorithm. In our study, we categorized our cases based on the value of the PASS score and analyzed additional prognostic factors. In numerous studies, angiogenesis has been recognized as a critical factor for the proliferation of tumors and the development of metastases [19]. Given the highly vascular nature of paragangliomas as a solid tumor, there has been interest in assessing the significance of angiogenesis in its pathogenesis, and investigations were prompted. Gao X. et al. investigated CD31 as an angiogenic marker, by counting the number of CD31 positive vessels within the highest expressed tumor area (MVD), and the number was significantly higher in pheochromocytomas with PASS < 4 than in those with PASS ≥ 4. Gao X. et al. also detected that intratumoral hemorrhage was significantly higher in pheochromocytomas with PASS ≥ 4 than PASS < 4, which could also result in relatively low CD31 status in histologically low-grade tumors [16]. H. Ohji et al. counted CD34 labeled blood vessels (MVD) and found no statistical association between the MVD and malignancy, while Q. Liu et al., who assessed MVD by staining endothelial cells with antibody to factor VIII/von Willebrand factor antigen, found a highly significant difference between the groups of benign and malignant [20,21]. In the former study, the cases were divided into benign and malignant according to the presence of metastatic disease, and in the latter group, besides the presence of metastasis, tumors with evidence of capsular or vascular invasion were included, which are histological criteria included in PASS scoring scale. Q. Lui also noticed that areas of highest vascularization were usually along the periphery of the tumor [20]. M. Bialas et al. studied angiogenesis status in pheochromocytomas, including MVD and vascular architecture, after immunostaining endothelial cells with antibodies CD31 and CD105 [33]. Vascular architecture patterns were highly heterogeneous within pheochromocytomas, both benign and malignant. The MVD in the central areas of the tumor was higher than in the subcapsular areas. Secondary changes in tumors, like hemorrhage and cystic degeneration, influenced the counting of MVD and assessing vascular architecture [33]. Favier et al. used immunostaining for CD34 and α-actin smooth muscle cells to define vascular architecture, and they reported differences, noticing two different patterns, mainly benign pheochromocytomas having more regular patterns, with short and straight vascular segments, while malignant presented with an irregular pattern, longer vascular segments, of irregular length [34]. The density of vascular segments was lower in irregular patterns, but blood vessel counting was not considered appropriate for these tumors [34]. L. Oudiijk et al. found that the mean sensitivity and specificity of vascular architecture as a predictor of potential malignant behavior of paragangliomas was 59,7% and 72,9%, with a significant agreement between the observers [35]. To assess tumor angiogenesis and to minimize the potential influence of marker choice on our results we chose to evaluate intratumoral MVD using several IHC markers at the same time. By making TMA we tried to select areas of the tumor with a high density of blood vessels without hemorrhages and cystic degeneration. While most studies, as was mentioned, have utilized CD31 and CD34, which are pan-endothelial cell markers that react with both proliferating and pre-existing vessels in tumors, CD105 (endoglin) is an endothelial antibody that primarily binds to activated endothelial cells in angiogenesis. Thus, it may be a more specific marker for tumor angiogenesis, as indicated by prior research [36,37,38,39]. In addition to the established antibodies used to measure intratumoral MVD, we examined ERG, a transcription factor belonging to the ETS family that is expressed by endothelial cells involved in angiogenesis. This approach has not been previously explored, to the best of our knowledge, on paragangliomas [40]. All four antibodies used for intratumoral MVD indicated that these were well-vascularized tumors. There was no statistically significant difference in intratumoral MVD with regard to PASS potentially malignant behavior. However, all four endothelial antibodies showed lower intratumoral MVD in the group of malignant tumors, suggesting that cells may be able to overcome hypoxia. Hypoxia has been mentioned as essential for angiogenesis in paragangliomas and due to the pseudohypoxia signaling pathway, they are well-known for increased angiogenesis [41,42]. Although neoplastic angiogenesis allows tumor spread, it does not imply that these well-vascularized tumors will metastasize. Additional knowledge is needed to understand the crucial features that drive the metastasis of paraganglioma and the role of angiogenesis in this process. In light of the high replicative potential observed in many malignant tumors, several studies have focused on the proliferative Ki67 index [43,44,45,46]. However, it has been found that this index has low sensitivity when it comes to intra-adrenal and extra-adrenal paraganglioma despite its high specificity. Paragangliomas are slow-growing tumors, and most show very low Ki-67 labeling [18]. Proliferative index Ki67 has high specificity which implies poor sensitivity, with only 50% of malignant tumors having a score greater than 2–3%, so the cut-off value for malignancy is low [18,47]. Our study confirmed the low sensitivity of Ki67 in this context. Nevertheless, Ki67 is an immunohistochemical marker that should be measured, as high Ki67 values may warrant attention for other prognostic factors, such as genetic predisposition and careful consideration of significant histologic predictors of malignancy, such as tumor necrosis [18]. Paragangliomas consist of chief cells that are visible on H&E staining, as well as sustentacular cells, that often necessitate a specific immunohistochemical antibody. While their precise role in metastasis development is not yet clear, several studies, including our own, have demonstrated a decrease in the number of sustentacular cells in malignant tumors [31,43,48,49]. In some cases in our study, these cells were entirely absent. These tumors have the strongest genetic contribution with nearly 40% of them associated with mutations in one of the known susceptibility genes, including NF1 (neurofibromatosis type 1), VHL, RET (rearranged during transfection), EPAS1 (endothelial PAS domain containing protein 1), EGLN1 (egl-9 family hypoxia inducible factor 1), SDHA (succinate dehydrogenase complex subunit A), SDHB, SDHC (succinate dehydrogenase complex subunit C), SDHD (succinate dehydrogenase complex subunit D), SDHAF2 (succinate dehydrogenase complex assembly factor 2), FH (fumarate hydratase), TMEM127 (transmembrane protein 127), MAX (Myc associated factor X), MDH2 (malate dehydrogenase 2), GOT2 (glutamic-oxaloacetic transaminase 2), SLC25A11 (solute carrier family 25 member 11), DLST (dihydrolipoamide S-succinyltransferase), H3F3A (H3.3 histone A), DNMT3A (deoxyribonucleic acid methyltransferase 3 alpha), MET (MNNG HOS transforming gene), MERTK (mer-tyrosine kinase), and KIF1B (kinesin family member 1B). There are different types of mutations including germline only, germline and somatic, somatic and somatic mosaicism [3,4,5,50]. Germline mutations involving genes coding for succinate dehydrogenases (SDHx) are the most common genetic cause of paraganglioma, occurring in up to 25% of cases [51,52]. They are followed by genes VHL (4–10%), RET (1–5%), and NF1 (1–5%) [53]. Testing for the germline SDHB mutation in patients with paraganglioma is recommended by Clinical Practice Guidelines [6,54]. Immunohistochemically SDHB negative staining in tumors carries a high risk for the presence of SDHx mutations, whether in SDHB, SDHC, SDHD, or SDHA [14]. Loss of SDHB immunoreactivity in tumor cells with SDHx mutations is reported with 100% sensitivity and 84% specificity, with a positive predictive value of 92% and a negative predictive value of 100% [14,55]. In our study, the majority of cases retained immunopositivity to SDHB, and only one had a confirmed mutation. Since genetic testing was not available in all cases, and unfortunately still is not routinely conducted, immunohistochemistry helped to potentially rule out an SDHx mutation. It was found that the malignant paragangliomas were larger than the benign ones with a median tumor size of 61 mm and 44 mm, respectively. However, the precise cutoff value that could precisely predict malignant behavior is not established. Since they are all potentially malignant, a staging system is introduced, with size of 5 cm as cut-off value for T1 and T2 [22]. Tumor size and weight are routinely reported, which makes them available for risk stratification, but they are not considered independent parameters. Invasive features, like an invasion of periadrenal fatty tissue despite being more common in malignant tumors, were not significantly correlated regarding potential malignant behavior in some studies, but it is a criterion for T3 [15,18,22]. The prognosis of malignant paragangliomas remains poor with overall 5-year survival less than 50% [56]. Management of malignant tumors includes surgery, radiation therapy, or chemotherapy and several targeted therapies have been investigated, but currently, there are no curative therapy options [57]. 

Limitations: In our study, we had a large number of consecutive cases, but the majority were adrenal tumors and a small number were extra-adrenal tumors which are usually have the SDHB mutation [14,58]. Some investigations indicated that extra-adrenal paragangliomas also have a higher and more heterogeneous proliferative index Ki67 than intra-adrenal [59,60,61,62]. Genetic testing was not performed on all patients, so the angiogenesis was not correlated with the tumor’s background completely.

## 5. Conclusions

In the end, it seems that it all comes down to genetics and molecular testing, which is not surprising when these tumors have so strong genetic contributions, but it is discouraging because routine genetic testing in all health centers is not yet available. Eventually, genetic testing will complete our knowledge and clarify some doubts, but in everyday practice, it is possible to assess stratification risk with greater accuracy using accessible data like tumor size, histological parameters (included in PASS score), and additional immunohistochemical markers (proliferative index Ki67, S100, and SDHB). Precise assessment of intratumoral MVD on scanned images, with immunohistochemical labeled blood vessels may help in understanding tumor biology and add precision to risk assessment.

## Figures and Tables

**Figure 1 diagnostics-14-00849-f001:**
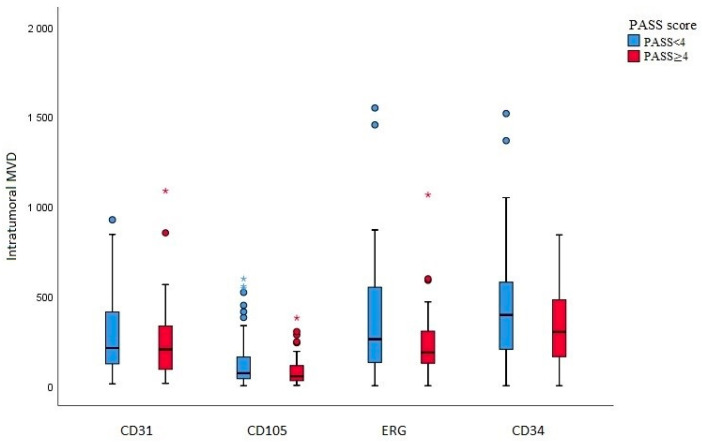
Box plot shows the number of blood vessels per mm^2^ (marked with antibodies CD31, CD105, ERG, and CD34) in relation to the PASS score (PASS < 4 and PASS ≥ 4), with some values labeled with * and °, that deviated extremely * and less extreme ° from the maximum.

**Figure 2 diagnostics-14-00849-f002:**
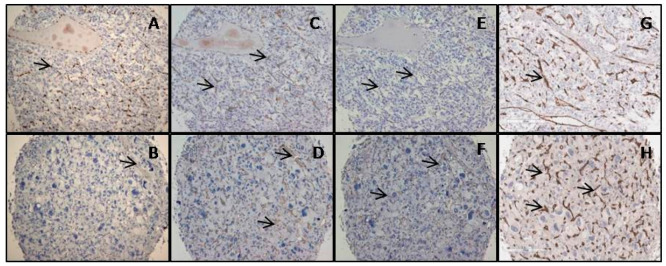
Intratumoral MVD immunostaining of blood vessels, in case of PASS < 4, labeled with ERG (**A**), CD31 (**C**), CD 105 (**E**), CD 34 (**G**), and PASS ≥ 4 labeled with ERG (**B**), CD31 (**D**), CD105 (**F**), and CD34 (**H**). Some blood vessels are indicated by the arrows. Magnification 200×, scale bar of 200 μm.

**Figure 3 diagnostics-14-00849-f003:**
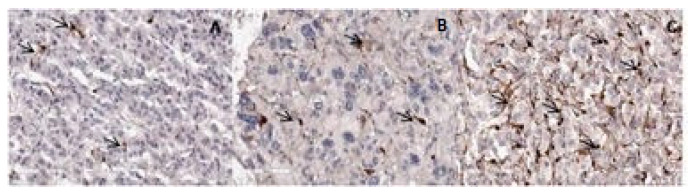
IHC staining showing S100 labeled sustentacular cells (arrows) in case evaluated PASS score 10 (**A**), PASS 6 (**B**), and PASS 3 (**C**). Magnification 400×, scale bar 90 µm.

**Figure 4 diagnostics-14-00849-f004:**
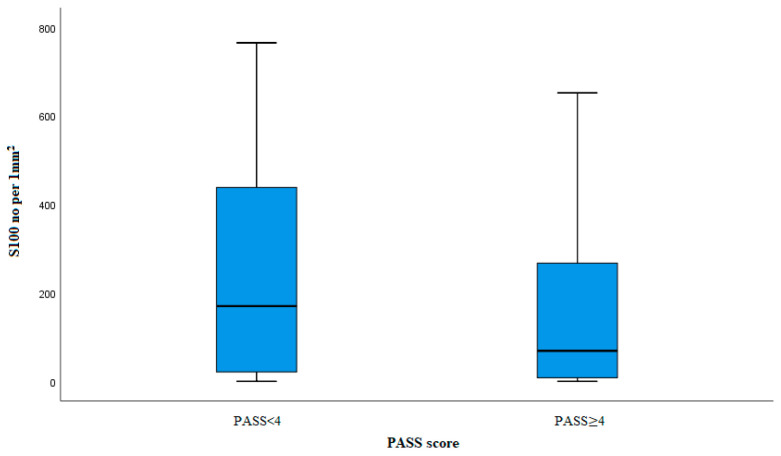
Box plot S100 labeled cells in groups PASS < 4 and PASS ≥ 4.

**Figure 5 diagnostics-14-00849-f005:**
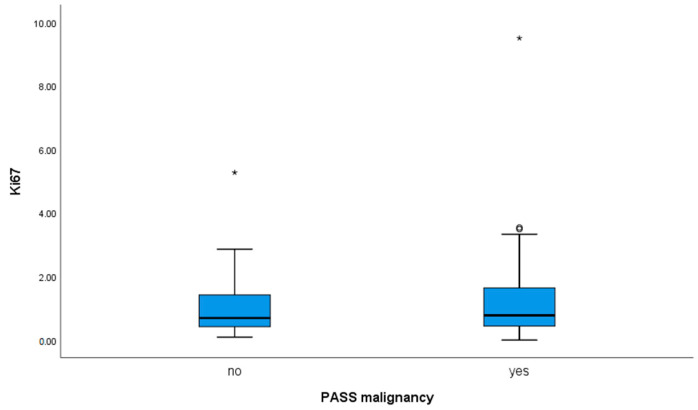
Box plot showing values of Ki67 in groups PASS < 4 and PASS ≥ 4, with labeled values which deviated extremely * and less extreme ° from the maximum.

**Figure 6 diagnostics-14-00849-f006:**
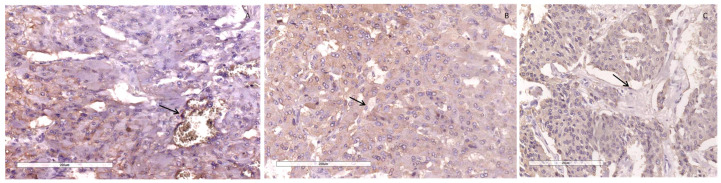
IHC SDHB lost expression in cases with PASS score 1 (**A**), PASS score 4 (**B**), and PASS score 8 (**C**). Some endothelial cells showing expressiveness are indicated by the arrows. Magnification 200×, scale bar 200 µm.

**Figure 7 diagnostics-14-00849-f007:**
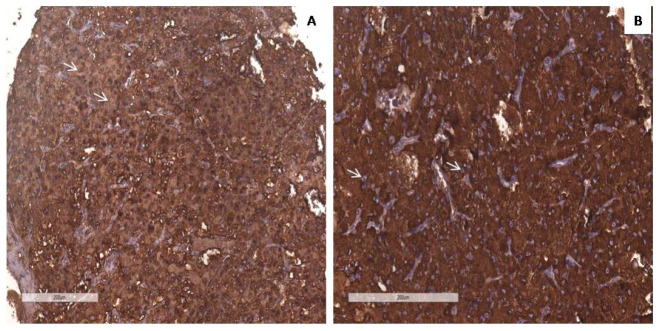
IHC SDHB positivity of tumor cells (arrows) in cases with PASS score 3 (**A**) and PASS score 9 (**B**). Magnification 200×, scale bar 200 µm.

**Table 1 diagnostics-14-00849-t001:** List of used antibodies and their manufacturers.

Antibody	Dilution	Source
CD31	1:100	BIO SB
CD34	1:200	NOVOCASTRA
CD105	1:100	Termo Fisher
ERG	1:100	ABCAM
SDHB	1:50	ABCAM
S-100	1:100	NOVOCASTRA
Ki67	1:200	DAKO

**Table 2 diagnostics-14-00849-t002:** Clinicopathological data in groups PASS < 4 and PASS ≥ 4.

Feature	PASS < 4	PASS ≥ 4	*p* ^e^
Total number (%) ^a^ (m, f) ^b^	61 (53%) (17 m, 44 f)	54 (47%) (23 m, 31 f)	
Age at resection mean/range	48.4 (22–78)	48.8 (19–73)	0.864
Site of tumor			0.118
Intraadrenal total/percentages	55 (47.83%)	52 (45.22%)	
extraadrenal total/percentages	6 (5.22%)	2 (1.74%)	
Tumor weight in grams mean/range	31.7 (3–159)	89.7 (2–361)	0.001
Tumor size in mm mean/range	44 (4–110)	61 (20–130)	0.001
Catecholamines/metabolites			0.673
elevated total/percentages	47 (40.87%)	46 (40%)	
normal total/percentages	9 (7.83%)	7 (6.08%)	
unknown data total/percentages	5 (4.35%)	1 (0.87%)	
Blood pressure/spikes			
elevated total/percentages	47 (40.87%)	44 (38.26%)	0.676
normal total/percentages	13 (11.3%)	10 (8.7%)	
unknown data total/percentages	1 (0.87%)		
MIBG scintigraphy			1.000
increased accumulation	24 (20.87%)	16 (13.91%)	
normal distribution	8 (6.96%)	4 (3.49%)	
unknown data	29 (25.22%)	34 (29.56%)	
Genetic testing			0.843
mutation present	16 (13.91%)	16 (13.91%)	
mutation excluded	8 (6.96%)	8 (6.09%)	
unknown data	37 (32.17%)	31 (26.96%)	
TNM			0.001
T1 tumor < 5 cm ^c^	37	13	
T2 tumor ≥ 5 cm ^d^	22	24	
T3 tumor of any size with invasion	2	16	

^a^ percentages, ^b^ male; female, ^c^ without invasion into surrounding tissues, ^d^ without invasion or sympathetic PG of any size, ^e^ *p* value, *p* < 0.05 considered significant.

**Table 3 diagnostics-14-00849-t003:** Examined IHC markers comparing with PASS score.

Antibody	PASS < 4	PASS ≥ 4	*p* ^b^
CD31 mean/SD ^a^	274.31/206.41	242.41/204.15	0.336
CD105 mean/SD ^a^	132.31/149.85	83.81/82.86	0.088
ERG mean/SD ^a^	358.87/321.36	234.46/179.92	0.071
CD34 mean/SD ^a^	429.28/324.73	319.50/226.88	0.077
S100 mean/SD ^a^	243.11/242.73	158.61/187.27	0.062
Ki67 mean/SD ^a^	1.02/0.93	1.22/1.47	0.598
SDHB positive/negative	57/4	52/2	0.683

^a^ SD Standard deviation. ^b^ *p* value, *p* < 0.05 considered significant.

**Table 4 diagnostics-14-00849-t004:** Results of genetic testing, age of patients, and PASS score values.

Syndrome	Number (m, f) ^x^	Age Mean/Range	PASS Mean/Range
VHL	10 (6 m, 4 f)	37.7 (19–55)	4.2 (1–12)
MEN2a	17 (6 m; 11 f)	42.6 (22–64)	3.1 (1–8)
NF1	4 (1 m; 3 f)	44.3 (34–53)	6.3 (4–9)
SDHB	1 (1 f)	51	9
Without mutation	15 (3 m; 11 f)	45 (29–64)	5.1 (1–10)
Not available	68 (24 m; 44 f)	52.5 (23–78)	3.7 (1–10)

^x^ m, male; f, female.

## Data Availability

The data presented in this study are available on request from the corresponding author.

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
