# Peer review of "Comprehensive Investigation of Angiogenesis, PASS Score and Immunohistochemical Factors in Risk Assessment of Malignancy for Paraganglioma and Pheochromocytoma"

_diagnostics, 2024, doi:10.3390/diagnostics14080849_

Round 1
Reviewer 1 Report
Comments and Suggestions for Authors
Manuscript Title: Comprehensive investigation of angiogenesis, PASS score and immunochistochemical factors in risk assessment of malignancy for Paraganglioma and Pheochromocytoma
. . . . . . . . . . . . . . . . . . . . . . . . . . . . . . . . . . . . . . . . . . . . . . . . . . . . . . . . . . . . . . . . . . . . . . . . . . . . . . . . . . . . . .
Comments on the current research paper on the investigation of angiogenesis, PASS score and immunocystochemical factors in malignancy risk assessment for paraganglioma and pheochromocytoma are as follows.
Presenting the methods discussed in the material and method section of the article under intermediate headings is important in terms of classification and will help the reader to follow the article under which title.
It is recommended that you include detailed explanations of some abbreviations in the article. For example, "MIBI scintigraphy", "MIBG scintigraphy". Apart from this, detailed explanations of some abbreviations were not found in the article.
It is recommended to define the name and unit of the "Y" axis in Figure 1.
It would be useful for the reader to indicate the numerical values of the microscope magnifications of the images in Figure 2. Additionally, using symbolic representation arrows for the measured blood vessel will provide similar benefits.
It would be useful for the reader to indicate the numerical values of the microscope magnifications of the images in Figure 3. Additionally, using symbolic representation arrows for the measured "sustentacular cells" will provide benefits.
Similar situations should be adopted for the images in Figures 6 and 7.
In addition to all these, Figure 5 was not found even though it was mentioned in the article. Writers need to compensate for this.
. . . . . . . . . . . . . . . . . . . . . . . . . . . . . . . . . . . . . . . . . . . . . . . . . . . . . . . . . . . . . . . . . . . . . . . . . . . . . . . . . . . . . .
Author Response
To Revier 1, in attachment, response on comments:
In the material and method section of the article we added intermediate headings.
We included explanation of abbreviations in the brackets.
We defined the name of the "Y" axis in Figure 1.
We added the numerical values of the microcopic magnification of the images in Figure 2 and Figure 3 and we used symbolic representation arrows for the measured blood vessels and for the mesured "sustentacular cells".
We added the numerical values of the microscope magnification for the images in Figure 6 and 7, and added arrows to represent marked cells.
We added Figure 5 on place mentioned in the article.

Reviewer 2 Report
Comments and Suggestions for Authors
This is a retrospective unicentric study of patients operated for paraganglioma/pheochromoctoma aiming to comparatively analyse angiogenesis by assessing intratumoral microvascular density (MVD) with immunohistochemical markers. Tissue samples were divided into two categories according to the PASS score. The problem studied is important, the study design is sound and the results clear.
The general comparison between subgroups presented in Table 2 needs to be accompanied by the statistical significance level. The same for the figures presented.
I suggest adding an additional table to include all data and statistical significance levels presented in the text, it would be much easier to follow.
Eventually no significant statistical difference in average intratumoral MVD in group PASS <4 and .4, actually MVD was lower in the potentially malignant group.The discussion is detailed and offers a nice perspective on the previously reported results: angiogenesis has been studied in these tumors and showed different conclusions regarding MVD and vascular architecture. Given the results of the current study and the heterogeneity of previously reported result I would rephrase the firm conclusion that assessment of intratumoral MVD is important.
Author Response
To Reviewer 2, in the attachment, we made changes as we understood:
The general comparison between subgroups presented in Table 2 is accompanied by the statistical significance level.
Additional table is added, which includes data and statistical significance levels for the main comparisons presented in the text.
We rephrased the firm conclusion that assessment of intratumoral MVD is important.